# The Regulation of Endoplasmic Reticulum Stress in Cancer: Special Focuses on Luteolin Patents

**DOI:** 10.3390/molecules27082471

**Published:** 2022-04-12

**Authors:** Roohi Mohi-ud-din, Reyaz Hassan Mir, Taha Umair Wani, Khalaf F. Alsharif, Waqas Alam, Ashraf Albrakati, Luciano Saso, Haroon Khan

**Affiliations:** 1Pharmacognosy & Phytochemistry Division, Department of Pharmaceutical Sciences, University of Kashmir, Hazratbal, Srinagar 190006, India; 2Pharmaceutical Chemistry Division, Department of Pharmaceutical Sciences, University of Kashmir, Hazratbal, Srinagar 190006, India; rehazhassan249@gmail.com; 3Pharmaceutics Division, Department of Pharmaceutical Sciences, University of Kashmir, Hazratbal, Srinagar 190006, India; tahaumair43@gmail.com; 4Department of Clinical Laboratory, College of Applied Medical Science, Taif University, P.O. Box 11099, Taif 21944, Saudi Arabia; alsharif@tu.edu.sa; 5Department of Pharmacy, Abdul Wali Khan University, Mardan 23200, Pakistan; waqasalamyousafzai@gmail.com; 6Department of Human Anatomy, College of Medicine, Taif University, P.O. Box 11099, Taif 21944, Saudi Arabia; a.albrakati@tu.edu.sa; 7Department of Physiology and Pharmacology, Sapienza University, 00158 Rome, Italy; luciano.saso@uniroma1.it

**Keywords:** cancer, ER stress, luteolin, luteolin combating ER stress, trigger proliferation of cancerous cells

## Abstract

Cancer is a major health problem across the globe, and is expeditiously growing at a faster rate worldwide. The endoplasmic reticulum (ER) is a membranous cell organelle having inextricable links in cellular homeostasis. Altering ER homeostasis initiates various signaling events known as the unfolded protein response (UPR). The basic purpose of the UPR is to reinstate the homeostasis; however, a continuous UPR can stimulate pathways of cell death, such as apoptosis. As a result, there is great perturbation to target particular signaling pathways of ER stress. Flavonoids have gained significant interest as a potential anticancer agent because of their considerable role in causing cytotoxicity of the cancerous cells. Luteolin, a flavonoid isolated from natural products, is a promising phytochemical used in the treatment of cancer. The current study is designed to review the different endoplasmic reticulum stress pathways involved in the cancer, mechanistic insights of luteolin as an anticancer agent in modulating ER stress, and the available luteolin patent formulations were also highlighted. The patents were selected on the basis of pre-clinical and/or clinical trials, and established antitumor effects using patent databases of FPO IP and Espacenet. The patented formulation of luteolin studied so far has shown promising anticancer potential against different cancer cell lines. However, further research is still required to determine the molecular targets of such bioactive molecules so that they can be used as anticancer drugs.

## 1. Introduction

One of the major health problems worldwide is cancer, which is a group of diseases caused by uncontrolled cell growth with invasive potentials and which metastasizes to other body parts. It is the second major cause of mortality globally following cardiovascular disease and is indeed a serious health issue in all human communities [1]. Researchers are greatly focusing in the chemoprevention area due to the increasing understanding of cancer biology, identification of molecular targets, and advancement in prevention of colon, prostate, and breast cancer [2]. Across the developed and developing countries, the incidence as well as the prevalence of cancer mortality is growing rapidly. As per New Galobocan 2018 cancer data, 18.1 million new cancer cases and 9.6 million cancer-related deaths have been documented from 185 countries [3]. Even though there is a remarkable decline in cancer mortality in the United States due to improvement in the diagnostic tools, advanced treatment approaches, and cancer awareness programs, the prevalence of cancer is still continuously growing. A large proportion of cancer incidences and deaths are because of various genetic and environmental factors, such as high body mass index, radiation exposure, heredity, lack of physical activity, alcohol intake, tobacco use, and low fruit and vegetable intake [4,5]. The incidence of cancer is marginally greater in men than women. The reasons behind the bias of cancer disease towards males are not well-known, however variation in environmental exposures and hormones may have a certain role [6,7].

Natural product-derived compounds could result in new, innovative therapeutic agents for cancer. Several promising new chemo-preventive agents based on their selective molecular targets have been developed and used in the clinic [8,9,10]. Besides, modern technology progress has enabled us to design drug molecules for particular molecular targets. Consequently, attention has been shifted towards natural origin agents from chemically synthetic drugs [11,12]. Flavonoids, secondary polyphenolic metabolites, have extensive pharmacological properties and have been evaluated against various diseases, including cancer [13], cardiovascular [14,15], anti-inflammatory [16,17], diabetes, hepatoprotective [18,19,20], antioxidant [21], and neurological disorders [22,23]. Flavonoids have gained significant interest as a potential anticancer agent because of their considerable role in causing cytotoxicity of the cancerous cells [24,25]. Among different types of flavonoids, luteolin (3′, 4′, 5, 7-tetrahydroxylflavone) is a common dietary form found in various medicinal herbs, vegetables, and fruits, such as carrots, parsley, broccoli, peppers, and celery [26,27]. It is mostly present in the form of glycosides in the plants, and is metabolized by intestinal bacteria. Luteolin has dynamic activity against cancer [28,29] and inflammation, and has been found to reverse multiple drug resistance (MDR) in various cancer cells [30,31]. The data available so far imply that luteolin has anticancer effects, including pro-apoptosis and arrest of the cell cycle, inhibiting metastasis and angiogenesis [26,32,33]. Moreover, it has been reported that by activating ER stress in neuroblastoma, luteolin can induce apoptosis [26]. The ER is the main site for synthesis, folding, and transport of proteins in the cell membrane, and of calcium signaling. It is well-known that handling of calcium (Ca^+2^) is one of the most extensive transducing systems, regulating extended cell biology and pathophysiologic processes [34,35]. Disruption of these pathophysiological functions results in the unfolded proteins’ accumulation and initiates the unfolded protein response [36].

This review aimed to focus on the overview of luteolin, as an anticancer agent, via modulating ER stress, involved mechanistic pathways, and the available patents disclosing luteolin as an anticancer agent. The inadequacy of anticancer drugs is mainly due to drug resistance. Newer drugs must have better pharmacological profiles. Here, we focus on luteolin as a therapeutic agent to target signaling pathways of ER stress.

## 2. Luteolin: An Overview

In recent years, natural product-derived molecules have gained significant interest within the scientific community due to their beneficial effects on human health in treating different diseases [37,38,39]. Luteolin is chemically 3′, 4′, 5, 7-tetrahydroxyflavones. It is an essential flavone, which is naturally present in many medicinal plants, vegetables, and fruits [40]. Structure–activity studies have shown that the presence of hydroxyl moieties at carbons 5, 7, 3′, and 4′ positions of the luteolin structure and the presence of the 2−3 double bond are responsible for its multiple pharmacological effects. Luteolin is a dynamic compound, characterized as a flavonoid with outstanding therapeutic potential, including anti-inflammatory, anticancer, antioxidant, and neuroprotective effects, which is largely documented in the literature [41,42,43,44,45,46].

## 3. A Brief Explanation of Its Pharmacological Activities

Luteolin is reported to possess antioxidant activity by increasing the Nrf2 activity that is responsible for its neuroprotective effect [47]. Further, the nephrotoxicity induced by lead acetate was prevented by luteolin by strengthening of the antioxidant and anti-inflammatory mechanisms by altering the Nrf2/HO-1 pathway [48]. In another study, luteolin exhibited antioxidant potential against various cancer cell lines [49].Luteolin possesses anti-inflammatory effects through the modulation of various inflammatory mediators in various in vitro and in vivo studies. Chen et al. authenticated the anti-inflammatory activity of luteolin in a mouse model. Suppression in the expression of the inflammation-associated gene was observed in mice due to the inhibition of NF-kappa B and AP-1 pathways in alveolar macrophages [50]. Administration of 10 and 50 mg/kg of luteolin orally suppressed the paw edema in carrageenan and prevented the leukocyte infiltration. Furthermore, downregulation in the COX-2 expression was observed in inflammatory conditions [51]. Luteolin has been demonstrated to suppress TNF-α expression in lipopolysaccharides induced in neonatal rat cardiac myocytes. The effect was observed through inhibition of the NF-κB signaling cascade [52].

Luteolin exhibits anticancer potential against the MCF-7 cell line through apoptosis-mediated cell death by activation of both the intrinsic and extrinsic signaling cascade, and also prompts cell cycle arrest [53]. Kumar et al. evaluated the anticancer potential of luteolin against colon cancer induced by administration of Azoxymethane and authenticated its preventive role in colon carcinogenesis through alteration of the Wnt signaling cascade by inhibiting cyclin D1and (GSK)-3β [54]. The anticancer potential of luteolin in small-cell lung cancer was evaluated and revealed fruitful results due to its pro-apoptotic and cell cycle arresting properties in the A549 cell line. Activation of JNK and hindering of translocation of NF-κB (p65) is responsible for its anticancer property [55]. Zang et al. [56] evaluated the anticancer potential of luteolin and confirmed its activity in KK-A(y) mice by suppressing the glucose levels. Luteolin decreases insulin resistance and improves insulin production and action through activation of the PPARγ pathway [57]. Luteolin exhibited a neuroprotective property against spinal cord ischemia-reperfusion injury in a rat model through the suppression of oxidative stress, inflammatory response, and apoptotic pathway. The possible mechanism might involve activation of Nrf2 and NLRP3 inflammatory pathways [58]. Liu et al. [59] authenticated the protective effect of luteolin in high-fat diet-induced cognitive impairment in C57BL/6J obese mice. Another study also showed the preventive role of luteolin in Parkinson’s disease through the inhibition of inflammation-mediated injury of dopaminergic neurons exhibited through inhibiting the activation of microglial cells [60]. Luteolin showed a preventive effect against ischemic stroke through a decrease in oxidative stress and apoptosis and increasing the expressions of claudin-5 [61].

Luteolin displayed a preventive role in myocardial injury through enhancement in the antioxidant defense system by stimulating the Akt and ERK signaling cascade, which subsequently triggered the activation of Nrf2 and increased heme oxygenase-1 (HO-1) expression [62]. Liao et al. demonstrated the protective effect in ischemia-reperfusion-induced myocardium injury in a rat model by decreasing the production of NO [63]. Luteolin derived from *Achillea millefolium* exhibited a protective effect against CCl_4_-induced hepatotoxicity. Administration of luteolin at a dose of 250 and 500 mg/kg restores the various biochemical parameters to normal. The immunomodulatory, antioxidant, and anti-inflammatory activity of luteolin were found to be involved in its hepatoprotective effect [64]. Pre-administration of luteolin through the intraperitoneal route at varied doses showed a preventive role in liver failure induced by d-galactosamineor lipopolysaccharide in a mouse model due to the suppression of apoptotic signaling pathways (both extrinsic and intrinsic) [65]. Another study revealed the effective role of luteolin in decreasing ethanol-induced hepatic steatosis [66]. The improvement in NAFLD in db/db mice was also known to occur after treatment with luteolin [67]. Luteolin demonstrated an antibacterial property against *Staphylococcus aureus*. The activity is exhibited through restricting the activity of topoisomerase 1 and 2 that are involved in nucleic acid and protein synthesis in humans [68]. Luteolin inhibited the growth of bacteria present in dental plaque [69]. Luteolin has shown fruitful results in exhibiting the antiviral activity against *Japanese encephalitis* virus (JEV), which is responsible for the development of the very fatal disease, viral encephalitis [70]. Luteolin halts the replication of dengue virus by inhibiting proprotein convertase furin [71]. Another study revealed the activity of luteolin against Herpes Simplex Virus Type 2 [72].

## 4. Endoplasmic Reticulum Stress: An Overview of Molecular Pathways

The endoplasmic reticulum is comprised of a complex membranous network that plays an essential role in the normal functioning of cells, particularly in the folding and synthesis of secretory and membrane proteins along the ER membrane. Disruption in the functioning of the ER due to various obnoxious stress stimuli results in the build-up of unfolded and misfolded proteins within the lumen of the ER, a cellular condition known as ER stress, which is associated with numerous diseases including cancer [73,74,75,76,77,78]. The cells counter ER stress via various adaptive mechanisms, to restore the protein homeostasis, termed as UPR. The unfolded or misfolded proteins accumulated in the ER are detected by three ER transmembrane sensors, including IRE1α, ATF6, and PERK (Figure 1), to overcome the stress [79]. Under normal circumstances, GRP78 or BiP, an ER resident chaperone, is bound to these three sensors and keeps them in an inactive state. However, once uncoupled from BiP, PERK and IRE1 result in the formation of oligomers or homodimers and actuate their downstream pathways via autophosphorylation, whereas ATF6 in active form shifts towards the nucleus and stimulates transcription of various ER chaperones [80].

### 4.1. IRE1 Pathway

The IRE1 pathway is an important arm of UPR that activates in both physiological and pathological conditions, such as protein secretion, ERAD, and lipid synthesis [81,82,83]. IRE1 is a type I receptor protein having a cytosolic C-terminus domain surrounding the Ser/Thr kinase domain, as well as the endoribonuclease domain and the N-terminal ER domain. IRE exists in two isoforms, IRE1-α and IRE1-β. IRE1-α is expressed on the ER membranes, whereas IRE1-β in the gastrointestinal tract. Under stress conditions of the ER, IRE1-α detaches itself from BiP, dimerizes, and then auto-phosphorylates to its active form, triggering endoRNase activation. This results in mRNA splicing of XBP1 protein, which encodes specific genes intricated in pro-survival responses. Moreover, IRE1α, during its active state, activates RIDD, which in turn triggers apoptosis. IRE1α also links with TRAF2, which leads to augmentation of ASK1 and JNK, and instigates apoptosis [75,84].

### 4.2. PERK Pathway

PERK, which is structurally alike to IRE1, is a type 1 ER transmembrane protein and in mainly expressed in secretory cells [85,86]. During UPR activation, PERK is activated by autophosphorylation and oligomerization, resulting in the attenuation of mRNA translation and the blockade of entry of new proteins in the ER [80]. The process involved phosphorylation conciliated eIF2α inactivation, which results in the blockade of protein synthesis, and thus decreases the protein burden of the ER during stress. Phosphorylation of eIF2α, in the meantime, manages the ATF4expression, which in turn imparts a vital role in the regulation of genes (pro-apoptotic C/EBP homologous protein) maintaining cellular homeostasis. ATF4 along with CHOP dephosphorylates eIF2α by upregulating the transcription of the GADD34 protein and growth arrest. ATF4-CHOP upon activation induces the apoptotic pathway, if ER stress is irreversible [87,88,89,90].

### 4.3. ATF6 Pathway

ATF6 is an ER transmembrane protein that exists in two forms, ATF6α and ATF6β, in mammalian cells [91,92]. ATF6α shifts to the Golgi apparatus upon UPR activation, where proteases (S1P and S2P) transform it into cleaved ATF6α (active form). In its active form, ATF6α translocate towards the nucleus, where it regulates the expression of genes. Besides, it also elevates the expression of CHOP, associated with cell death [93,94].

## 5. Endoplasmic Reticulum Stress in Organ Damage

Endoplasmic stress is an adaptive response while dealing with misfolded or unfolded proteins to maintain homeostasis in the lumen of the ER. However, prolonged ER stress causes cell death, mostly via apoptosis. The process of apoptosis in the long run causes a disturbance of normal physiological functions and even damage to various organs of the body, resulting in various diseases, including diabetes, cardiovascular, and liver disease.

### 5.1. ER Stress and Cardiovascular Diseases (CVD)

The heart is very sensitive to ER stress. ER stress along with its UPR pathways play a very important role in the progression of CVD, including heart failure, ischemic heart disease, and atherosclerosis [95,96]. Long-term ER stress via exogenous or endogenous factors may result in the production of primary ROS, which leads to various CVD. Primary ROS includes superoxide anions that degrade lipids or proteins within the cell. ER stress may elevate the production of ROS by activating NADPH oxidase (Nox), particularly Nox-2 and Nox-4, upregulation of protein kinase JNK through IRE1, and folding and refolding of proteins in the lumen of the ER, which are factors mediating the release of ROS, particularly superoxide anion, in the CVS. Afterwards, superoxide anion integrates with NO to form peroxynitrite, resulting in endothelial dysfunction, an important hallmark for CVD. Primary ROS generated during ER stress in turn triggers various secondary ROS, including hypochlorous acid, hydroxyl radical, hydrogen peroxide, and hydroperoxyl radical, which aggravates further advancement of CVD [96,97,98,99,100,101,102,103]. Inflammation is also linked with ER stress and imparts an essential role in CVD [104]. Upon UPR activation, IRE-1 raises the level of TRAF2, which along with IκB kinase and JNK upregulates NF-κB. The NF-κB dimers released from the cytoplasm increase the inflammatory cytokines expression. Besides, activation of IRE1α promotes NLRP3 signaling, which stimulates IL-1β secretion for inflammation [105,106,107,108]. If the UPR pathway fails to reduce ER stress in the CVS, extrinsic and intrinsic pathways of apoptosis are upregulated [109]. The cells adopt various apoptotic pathways, which include PERK, CHOP, IRE1-mediated upregulation of TRAF2 stimulating apoptosis signal-regulating kinase, and Bax/Bcl-2-mediated Ca^2+^ release from the ER [110].

### 5.2. ER Stress and Diabetes

Diabetes involves complex physiological changes, which include dysregulated hepatic glucose production, inadequate insulin secretion, and peripheral IR. Various in vitro and in vivo studies confirm that in both type 1 and type 2 diabetes, ER stress induces pancreatic destruction of βcells, even though type 1 and type 2 diabetes have different etiologies as well as triggers [111]. In type 2 diabetes, there is persistent hyperglycemia as β cells are unable to produce insulin, or there is a development of insulin resistance by various tissues. The ER is the main site involved in the processing, synthesis, as well as storage of proinsulin. The translation rate of proinsulin synthesis in pancreatic βcells increases during elevated blood glucose levels, resulting in misfolding of proinsulin and β cells under ERS [112,113,114,115]. In order to eliminate the misfolded proinsulin, the UPR signaling pathway activates. However, due to continuous hyperglycemia, the defensive UPR pathway becomes devastating for the β cells. Among the UPR sensors, the PERK pathway has an essential contribution in regulating the development of β cells, homeostasis, and proliferation. This is supported by studies [116,117,118,119] showing that PERK-deficient mice manifest severe β cell destruction, followed by diabetes. PERK was also reported to be involved in governing the proinsulin via maintaining the ER chaperones’ expression. Under normal physiological conditions, IRE1a increases the biosynthesis of proinsulin after a meal. However, continuous high glucose levels lead to ERS-induced overactivation of the IRE1a pathway, resulting in reduced insulin genes’ expression in β cells. Tsuchiya et al., [120] via the IRE1a-XBP1 pathway, found that in mouse β cells, deficiency of IRE1a leads to diabetes, with the decreased folding of proinsulin and insulin. Therefore, overall, homeostasis of β cells depends on the balance between IRE1a and PERK activity, where IRE1a regulates insulin biosynthesis positively and PERK regulates it negatively. Any imbalance can be detrimental to β cells. The role of ATF6 in diabetes is limited. Thus, ERS as well as UPR components act as a new approach in the treatment of diabetes [116,117,118,119,120,121,122,123].

### 5.3. ER Stress and Lipid Metabolism

Within the cell, sterols and phospholipids are synthesized in the ER and are the lipid constituents of all membranes. However, during obesity, extra energy is stored in adipocytes inside the lipid droplets as triglycerides [124]. Cells go through extreme stress to consolidate more proteins for the generation of lipid droplets and maintain glucose deprivation and energy expenditure for metabolism during lipotoxicity, which is a main cause of cellular damage during pathophysiological conditions [125]. This depicts the role of the ER in lipid droplets’ formation to maintain the metabolism of lipids [126]. However, the formation of excessive proteins along with their transportation inside the ER commences the UPR signaling [127]. Various studies report that UPR plays an essential role in lipid and metabolic homeostasis. ATF6, PERK, and IRE1 are ER stress-responsive proteins and impart an essential role in the metabolism of lipids. The IRE1-XBP1 pathway is the main regulator of hepatic lipid metabolism. Deletion of IRE1 in liver cells upregulates hepatic lipid levels by elevating the expression of genes, including CCAAT binding protein as well as genes involved in the synthesis of triglycerides. Moreover, silencing of XBP1 downregulates cholesterol and FFAs by decreasing lipogenic genes such as ACC and SCD. The ATF6 pathway also has an essential role in the metabolism of lipids. It regulates the metabolism of lipids via a negative interaction with SREBPs and decreases the accumulation of lipids in both kidney as well as liver. ER stress and the accumulation of lipids in kidneys is mainly due to SREBP-2. The PERK pathway also imparts a critical role in adipocytes’ differentiation and adipogenesis by activating SREBP-1, and eIF2α results in inhibition of translation. Moreover, PERK enhances the translation of SREBP-1, ATF4, and mRNAs of GRP-78 [128,129,130,131,132,133,134,135].

## 6. ER Stress in Cancer Therapy

The ER is responsible for mediating a vast number of cellular processes through the synthesis, storage, and secretion of functional proteins in a cell [136]. This is the site for folding and synthesis of proteins, which represent a major proportion of proteins manufactured in a cell. However, certain detrimental situations such as increased protein secretary load or production of mutated proteins do not allow these proteins to fold correctly, and hence lead to the aggregation of unfolded proteins in the ER [137,138]. When very high levels of such unfolded proteins are produced by a cell, it results in an increase in the physiological stress on the ER, known as ER stress. Consequently, a mechanism to deal with this imbalance in the cellular homeostasis is activated, which is known as UPR. During UPR, such high levels of unfolded proteins are detected by certain sensor proteins, such asIRE1 (inositol-requiring protein) [139], ATF-6 (activating transcription factor) [140], and PERK [141]. A schematic representation of the above-discussed processes is shown in Figure 2. These processes stimulate an adaptive pro-survival mechanism and increase the production of ER to reinforce cell survival. Many times, even the increased number of ER is not enough to deal with such an overwhelming production of unfolded proteins, and this could lead to the secretion of these defective proteins and reach numerous destinations to cause various devastating problems. In such an uncontrolled situation, UPR stimulates a pro-apoptotic response, which ultimately triggers cell death that can lead to conditions such as diabetes and neurodegenerative disorders if cells carrying vital processes are affected. An interesting fact about cancer cells is that these cells use this UPR for appropriating the production of excessive amounts of ER for speedy proliferation of tumors [142]. Thus, UPR and aggravated production of ER is a normal process carried out in cancer cells for their excessive growth and uncontrolled proliferation. Researchers have looked upon this as an opportunity for targeted chemotherapy to aggravate UPR and increase the ER stress up to the limit where pro-apoptotic stimulus is triggered. Convincing evidence of this possibility has already been accumulated by researchers, where ER stress has been targeted for successful chemotherapy.

## 7. Luteolin and Endoplasmic Reticulum Stress

Among the vast number of mechanisms of the anticancer activity of luteolin is the cell apoptosis through ER stress and UPR, for which there is substantial amount of evidence in the literature. Kim et al. confirmed the induction of ER stress being involved in the cytotoxic effect of luteolin through increasing ROS and Ca^2+^ in mitochondria. The florescence microscopy showed very high intensity of the ER tracker dye in cells treated with luteolin, contrary to the control cells. The decrease in the cell proliferation was also revealed by the colony-forming assay, where it was found that the apoptotic bodies’ formation was increased in luteolin-treated cells as compared to the untreated control cells. The expression of the proteins associated with the ER stress and also the reactive oxygen species levels were elevated in the melanoma cells after treatment with luteolin. Another very important assay, the Western blotting assay, was used to estimate the proteins associated with ER stress, and proteins such asATF6, eIF2α, cleaved caspase 12, PERK, and CHOP were found to be present in higher proportions in luteolin-treated cells [143].

Another study was conducted by Park et al. on lung carcinoma cells to elucidate the mechanism of anticancer activity of luteolin. According to these authors, the luteolin-induced cell death occurs through ER stress and non-canonical autophagy, as evaluated in NCI-H460 cells. Assays such as flow cytometry, DNA fragmentation assay, Western blotting, etc., were used to for DNA and various ER proteins. The chromatin condensation, DNA fragmentation, and apoptotic bodies, indicative of cell death, were observed in luteolin-treated cells, but not the increase in G1 stage cells. Among the various other mechanisms, apoptosis was also shown to be mediated through ER stress. While investigating this, it was found that luteolin increases ER stress expression-related proteins such as CHOP and p-eIF2α. As the cells were treated with eIF2α siRNA, the expression of CHOP and p-eIF2α was remarkably reduced, which prevented the apoptotic activity of luteolin, clearly suggesting the involvement of ER stress in apoptosis after luteolin treatment [144]. This is demonstrated in Figure 3.

In yet another study, Lee et al. showed that luteolin exerts apoptotic activity in hepatocellular carcinoma cells via ER stress-activated p53 and decreases the number of cells. p53 is a tumor protein that functions to suppress gene mutations and thus prevent tumor formation. These authors evaluated two types of cells, p53-wild and p53-null cells, to establish whether ER stress is a relevant target for suppressing the growth of hepatocellular cancer with luteolin. The immunoblotting and PCR carried out on both types of cells revealed that the p53-nullcells showed significant catalase downregulation and mRNA upregulation. This meant that there was respectively an induction of oxidative stress and ER stress in luteolin-treated cells [145].

### 7.1. Anticancer Mechanism of Luteolin, Combating ER Stress

Luteolin is a flavonoid that has been shown by numerous researchers to induce cytotoxicity in several types of cancer, such as breast, lung, colon, liver, pancreatic, brain, kidney, prostate, ovarian, skin, oral cancers, etc. It has been reported to exert its anticancer action by inhibiting the proliferation of cancer cells, raising the amount of ROS, inhibiting carcinogenic stimulation, and by bringing about cell death through ER stress [29]. Luteolin acts on multiple signaling pathways in different cancers that mediate the tumor proliferation. Luteolin inhibits the metabolism of carcinogens and mutagen activation in cancer cells through inhibition of CYP450 enzymes, for example in liver microsomes [146,147,148]. Chowdhury et al. have demonstrated in their study that luteolin acts on DNA topoisomerase I in eukaryotic cells, preventing the formation of enzyme-DNA complex and entirely inhibiting its catalytic activity [149]. Luteolin also acts on protein kinases that are actively involved in the cell cycle regulation and growth of the tumors, and inhibits their activity. One such protein kinase is VRK1 (Vaccinia-related kinase 1). This protein kinase regulates the functioning of various cell cycle factors such as histone H3, BAF, and CREB through phosphorylation [150]. Luteolin has also been shown to impart its anti-tumor activity through cell cycle arrest as well. This has been demonstrated by Zhao et al. in A549 lung cancer cells, where they found that luteolin increased the number of cells in the G1 phase of their mitotic cycle while decreasing the S and G2 or M phase cells. Luteolin was also reported to obliterate the actin cytoskeleton and suppress the development of stress fiber, which ultimately led to restraining the migration of the cancer cells [151]. In addition to this, luteolin also facilitates the Bad and Bax upregulation (the pro-apoptotic proteins) and Bcl-2downregulation (an anti-apoptotic protein). It also causes mitochondrial dysfunction by carrying Bad and Bax from cytosol to mitochondria and cytochrome c out into the cytosol, which triggers apoptosis. Luteolin is also believed to induce endonuclease activation [152] in the cytoplasm by increasing the levels of caspase [153] and subsequently the proteolytic activity, which ultimately results in degradation of nuclear contents and cell death.

### 7.2. Available Patents of Luteolin Formulations as an Anticancer Agent

Advanced patent databases of FPO IP and Espacenet were used to study the available patent formulations of luteolin. The patents were selected on the basis of pre-clinical and/or clinical trials, and established antitumor effects. Four patents were identified, three from the USA and one from Europe. A patented luteolin pharmaceutical composition from the USA has reported that the luteolin formulation prevents and treats the cancerous cells of liver, brain, pancreas, lung, breast cancer, and leukemia via targeting the BCL-2, bromodomains, sirtuins, kinases, and matrix metalloproteinases [154]. Cohen invented a patent for the pharmaceutical formulation of luteolin (BZL101). He performed clinical trials on different cancer cells and reported that luteolin is useful in the treatment of anal cancer, endometrial cancer, gallbladder cancer, Hodgkin’s disease, nasopharyngeal cancer, prostate cancer, osteosarcoma, vaginal cancer, testicular cancer, Kaposi’s sarcoma, esophagus cancer, thyroid cancer, ovarian cancer, aplastic anemia, stomach cancer, breast cancer, etc. [155]. The composition of Cohen’s patent formulation is presentedin Table 1.

Kalvin and his team invented a potent novel pharmaceutical composition of TEGAFUR (chemotherapeutic agent), luteolin, catechin, and indole-3-carbinol (Table 2). They reported that the formulation was useful in vivo and clinically in the management of malignant carcinomas [155]. In this review, all patents confirmed having an excellent anticancer effect for numerous kinds of cancer cells. Moreover, the inventors suggested further pre-clinical and clinical trials for clarity and precise indication of reviewed patents and luteolin products.

Traditionally, in Chinese medicine, luteolin-containing herbs areused for the management of inflammation, cancer, and hypertension. A patent for a nutraceutical formulation containing luteolin under the trade name ProFine^TM^ is used in the treatment of different cancer cell lines of neck, head, and prostate cancer. The composition of a1 mL oral dose of ProFine^TM^ is presented in Table 3 [40].

## 8. Conclusions

The ER is considered as a varied signaling organelle that is involved in various biological processes, such as Ca^2+^ signaling, folding of proteins, and evolutionarily signaling pathways, known as the UPR during various levels of ER stress initiated by several stimuli. Unrestrained ER stress may result in the pathogenesis of several diseases. Therefore, in the treatment of various diseases, regulation of ER stress could be an effective therapeutic strategy. For this, the molecular mechanism of the UPR pathway for each disease is indispensable, and is still unknown. The current review described how UPR along with its components maintain metabolic and homeostatic processes. The goal of this review was to provide all possible information related to ER stress and its involvement in health functions. Plant-derived phytoconstituents are well-known for their preventive role against various diseases, including cancer, and this is well-documented in the literature. Furthermore, these phytochemicals can also be used as an alternative medicine against various human cancers and are considered safe and more effective against various cancers. In this context, luteolin, a flavonoid found in various vegetables and fruits, has been known for its anticancer activity by various mechanisms, including cell cycle arrest, inhibiting angiogenesis, metastasis in various cancer cell lines, and apoptosis. The data disseminate the cross-talks between luteolin and selected ER response biomarkers (UPR).

In this review, patent formulations of luteolin were discussed which are proven to have an anticancer effect both in vivo and in clinical trials. However, more clinical work needs to be carried out on various pharmacokinetic parameters by involving further human subjects before luteolin becomes a prescription drug. In modern times, a variety of anticancer compounds for various signaling of cancer cells have been established and utilized clinically for the production of novel anticancer drugs. Moreover, a patent of a luteolin pharmaceutical composition from the USA has treated and prevented the cancerous cells of liver, brain, pancreas, lung, breast cancer, and leukemia via targeting the BCL-2, bromodomains, sirtuins, kinases, and matrix metalloproteinases. A patent of a pharmaceutical formulation of luteolin (BZL101) invented by Cohen established that luteolin is useful in the treatment of anal cancer, prostate cancer, osteosarcoma, vaginal cancer, endometrial cancer, gallbladder cancer, Hodgkin’s disease, ovarian cancer, aplastic anemia, stomach cancer, breast cancer, nasopharyngeal cancer, testicular cancer, Kaposi’s sarcoma, esophagus cancer, thyroid cancer, etc. Furthermore, a potent novel pharmaceutical composition of TEGAFUR (chemotherapeutic agent), luteolin, myricetin, kaempferol, catechin, and indole-3-carbinol was found effective in the treatment of malignant carcinomas.

This review has summarized the role of luteolin in the endoplasmic stress for the management of cancer, molecular pathways involved, and the available patent sources of luteolin in the prevention and treatment of cancerous cells in different cancer cell lines. Numerous industries have started clinical studies, either on their own or in conjunction with certain molecular targeting pathways. However, further research is still required to determine the molecular targets of such bioactive molecules so that they can be used as anticancer drugs.

## Figures and Tables

**Figure 1 molecules-27-02471-f001:**
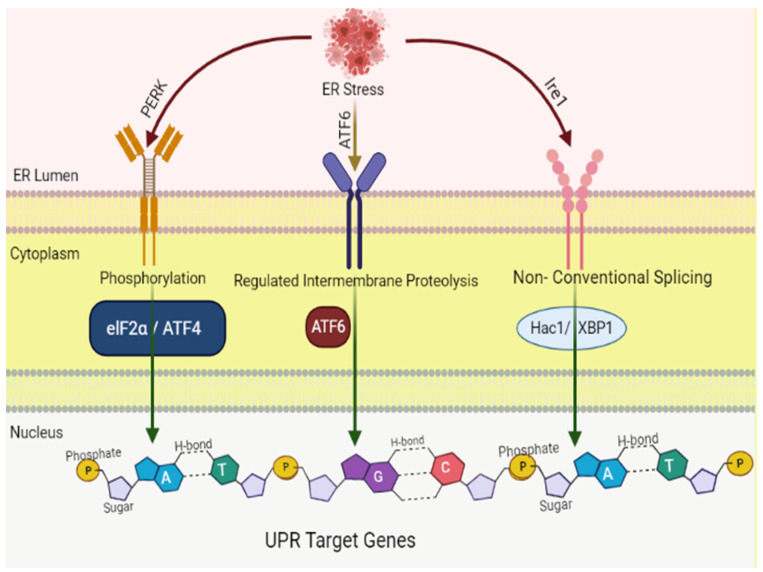
Signaling pathways initiated by the three branches of the unfolded protein response (UPR). Three signaling pathways are involved in the UPR, each of which is controlled by three ER membrane proteins: ATF6 (activating transcription factor 6), IRE-1 (inositol-requiring protein 1; official name: ERN1), and PERK (protein kinase-like ER kinase, pancreatic ER eukaryotic translation initiation factor (eIF)-2a kinase; official name: EIF2AK3). The UPR is a complicated cellular signaling mechanism that tries to reestablish cellular equilibrium. However, in the face of sustained ER stress, pathways that induce apoptosis may be triggered.

**Figure 2 molecules-27-02471-f002:**
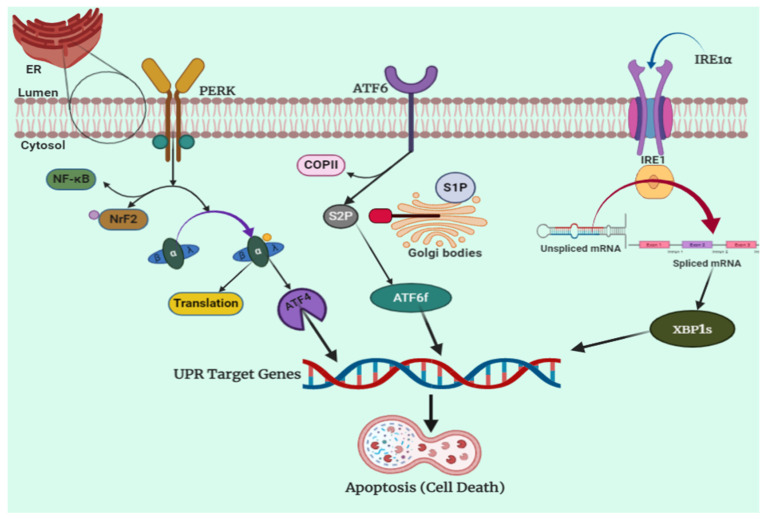
Endoplasmic reticulum stress pathway. The ER stress-associated unfolding protein accumulation stimulates UPR through sensor proteins such asIRE1, ATF-6, and PERK. The ER membrane contains three sensors that detect ER stress: PERK, ATF6, and IRE1. Grp78 dissociates from each of the unfolded proteins that accumulate in the ER lumen, prompting their stimulation. When PERK dimerizes and becomes autophosphorylated, its cytosolic kinase domain is activated, phosphorylating eIF2. Cap-dependent translation of ATF4 is enabled as a result of this suppression of overall protein synthesis. Grp78 dissociation induces ATF6 translocation to the Golgi, where it is translated by site 1 and site 2 proteases (S1P and S2P) into a functional transcription factor that activates XBP1 mRNA transcription. Grp78 dissociation from IRE1 causes IRE1 dimerization and autophosphorylation. When IRE1 is activated, its endoribonuclease activity causes un-spliced XBP1 mRNA to be processed, while its kinase site recruits TRAF2 and ASK1, causing JNK to be stimulated.

**Figure 3 molecules-27-02471-f003:**
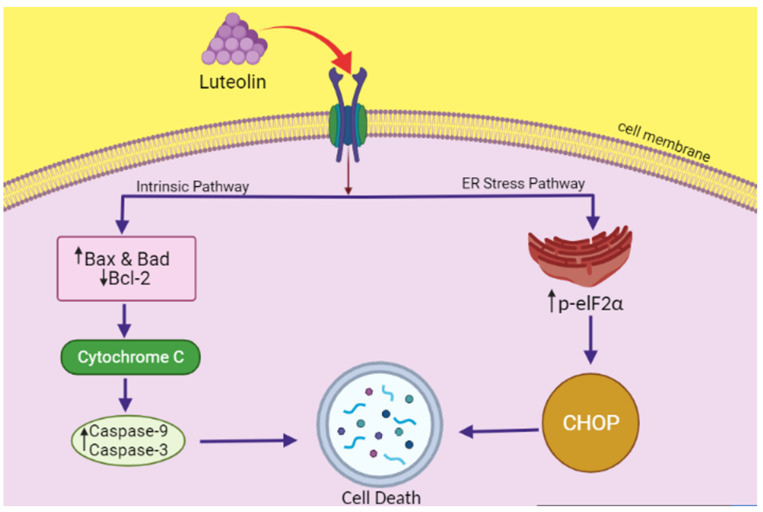
The schematic representation showing luteolin exerting its anticancer activity in NCI-H460 cells through the ER stress pathway. Luteolin increases ER stress expression-related proteins such as CHOP and p-eIF2α. As the cells were treated with eIF2α siRNA, the expression of CHOP and p-eIF2α was remarkably reduced, which prevented the apoptotic activity of luteolin, clearly suggesting the involvement of ER stress in apoptosis after luteolin treatment. Luteolin also stimulates Bax, Bad, and Bcl2, in turn activating caspase-3 and 9, leading to apoptosis (cell death).

**Table 1 molecules-27-02471-t001:** Composition of Cohen’s patent [155] luteolin formulation as an anticancer agent.

Composition	Quantity
Luteolin	1 Part
Apigenin	0.6–2 Parts
Scutellarein	2.5–9 Parts
Scutellarin	15–70 Parts

**Table 2 molecules-27-02471-t002:** Composition of the patent for TEGAFUR and chemotherapeutics agents.

TEGAFUR Capsules	
Composition	Quantity
Tegafur	100 mg
Luteolin	50 mg
(+)-Catechin	25 mg
Lactose	50 mg
Talc	50 mg
Indole-3-carbinol	50 mg
Ca stearate	7 mg
Aerosil	3 mg
**TEGAFUR Suspensions**	
Composition	Quantity
Tegafur	100 mg
Luteolin	50 mg
Glycerin	50 mg
Distilled water	100 mL
Indole-3-carbinol	50 mg
**TEGAFUR Suppositories**	
Composition	Quantity
Tegafur	300 mg
Luteolin	100 mg
Indole-3-carbinol	50 mg
Witepsol W-35	750mg
**TEGAFUR Tablets**	
Composition	Quantity
Tegafur	40 mg
Luteolin	5 mg
Crystalline cellulose	50 mg
Lactose	100 mg
Talc	10 mg
Indole-3-carbinol	10 mg
Magnesium stearate	10 mg

**Table 3 molecules-27-02471-t003:** Nutraceutical composition of ProFine^TM^.

Composition	Quantity
Luteolin	24.68 mg
Kaempferol	49.35 mg
Quercetin	26.06 mg
Corn oil	35%*v*/*v*
Ethanol	10%*v*/*v*
Tween 80	5%*v*/*v*
Hydroxylpropyl methylcellulose	50%*v*/*v*

## Data Availability

The data presented in this study are available upon request from the corresponding author.

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
