# Peer review of "The Regulation of Endoplasmic Reticulum Stress in Cancer: Special Focuses on Luteolin Patents"

_molecules, 2022, doi:10.3390/molecules27082471_

Round 1

Reviewer 1 Report

In the manuscript molecules-1636544, the authors reviewed the role of luteolin in the endoplasmic stress for the management of cancer, molecular pathways involved and the available patent sources of luteolin in the prevention and treatment of cancerous cells in different cancer cell lines. The patents formulation of luteolin showed promising anticancer potential against different cancer cell lines. It is an interesting and timely review. It is recommended for publication after addressing the below concerns

Comments

  • A Graphic Abstract is needed for the review to summarize the contents
  • For table 2, it should be rearranged for better reading.
  • The typos in the manuscript should be revised, calcium (Ca2p), detatches…
  • The literature is recommended to update. And the authors mentioned that natural products derived compounds could result in new, innovative therapeutic agents for cancer. Several promising new chemopreventive agents based on their selective molecular targets have been developed and used in the clinic. And it is recommended to cite some recent important report, J. Bioresour. Bioprod. 2020, 5(4): 223-237. J. Bioresour. Bioprod., 2021, 6(1): 26-32.

Author Response

On the behalf of all author, I am very grateful to the editor and reviewers for the valuable suggestions. We have positively address all of them and I am sure that their incorporation will greatly aid to the overall strength of our article.    

Author 1:

Comments and Suggestions for Authors

In the manuscript molecules-1636544, the authors reviewed the role of luteolin in the endoplasmic stress for the management of cancer, molecular pathways involved and the available patent sources of luteolin in the prevention and treatment of cancerous cells in different cancer cell lines. The patents formulation of luteolin showed promising anticancer potential against different cancer cell lines. It is an interesting and timely review. It is recommended for publication after addressing the below concerns

Comments

  1. A Graphic Abstract is needed for the review to summarize the contents

Response: The needful suggested changes have been done and highlighted.

  1. For table 2, it should be rearranged for better reading.

Response: The needful suggested changes have been done and highlighted.

  1. The typos in the manuscript should be revised, calcium (Ca2p), detatches…

Response: The needful suggested changes have been done and highlighted.

  1. The literature is recommended to update. And the authors mentioned that natural products derived compounds could result in new, innovative therapeutic agents for cancer. Several promising new chemopreventive agents based on their selective molecular targets have been developed and used in the clinic. And it is recommended to cite some recent important report, J. Bioresour. Bioprod. 2020, 5(4): 223-237. J. Bioresour. Bioprod., 2021, 6(1): 26-32.

Response: The suggested reports have been cited and highlighted.

Reviewer 2 Report

 The title looks interesting, but the content need to be arranged specifically with respect to Luteolin . Many general descriptions are included.

The manuscript entitled as

 “The regulation of Endoplasmic reticulum stress in Cancer: Special focuses on luteolin patents” is an interesting topic. But it needs a systematic rearrangement.

 I have observed a few concerns

1. The details of mechanistic insights of luteolin as anticancer agent in modulating ER stress and the available luteolin patent formulations need to be given much weightage rather than looking into general perspective of luteolin.

2. Pharmacological activities of luteolin can be divided as preclinical and clinical studies. The article will be more readable if the authors arrange it systematically

3. Another approach  is that the authors can modify  the draft such that relevance of ER in cancer therapy and role of flavanoids.

4. The review is not giving clear picture about the perspective of author view points, whether they need to highlight the luteolin preclinically and clinically orelse the the ER path way in cancer therapy. Some connections are missing in the draft , it is quiet confusing.

Author Response

On the behalf of all authors, I am very grateful to the editor and reviewers for the valuable suggestions. We have positively addressed all of them and I am sure that their incorporation will greatly aid to the overall strength of our article.    

Author 2:

Comments and Suggestions for Authors

 The title looks interesting, but the content need to be arranged specifically with respect to Luteolin . Many general descriptions are included. The manuscript entitled as “The regulation of Endoplasmic reticulum stress in Cancer: Special focuses on luteolin patents” is an interesting topic. But it needs a systematic rearrangement. I have observed a few concerns

  1. The details of mechanistic insights of luteolin as anticancer agent in modulating ER stress and the available luteolin patent formulations need to be given much weightage rather than looking into general perspective of luteolin.

Response: The needful suggested changes have been done and highlighted.

  1. Pharmacological activities of luteolin can be divided as preclinical and clinical studies. The article will be more readable if the authors arrange it systematically

Response: Thank you very much for your kind suggestions. The Author’s aim is not to provide the review of the preclinical and clinical studies of Luteolin.

  1. Another approach is that the authors can modify the draft such that relevance of ER in cancer therapy and role of flavanoids.

Response: Thank You very much for the valuable suggestions. The review is based on the role of luteolin in the regulation of endoplasmic reticulum stress in cancer

  1. The review is not giving clear picture about the perspective of author view points, whether they need to highlight the luteolin preclinically and clinically orelse the the ER path way in cancer therapy. Some connections are missing in the draft , it is quiet confusing.

Response: In this article, the author’s point of view is to summarize the endoplasmic reticulum stress in cancer and the role of luteolin in endoplasmic reticulum stress, and the available luteolin patents in cancer therapy. The Author’s aim is not to provide a review on the preclinical and clinical studies of Luteolin. There are no inclusion criteria for preclinical or clinical analysis.

Thanks

Prof. Dr. Haroon Khan

Round 2

Reviewer 2 Report

No comments